# Expression of Basement Membrane Molecules by Wharton Jelly Stem Cells (WJSC) in Full-Term Human Umbilical Cords, Cell Cultures and Microtissues

**DOI:** 10.3390/cells12040629

**Published:** 2023-02-15

**Authors:** David Sánchez-Porras, Daniel Durand-Herrera, Ramón Carmona, Cristina Blanco-Elices, Ingrid Garzón, Michela Pozzobon, Sebastián San Martín, Miguel Alaminos, Óscar Darío García-García, Jesús Chato-Astrain, Víctor Carriel

**Affiliations:** 1Tissue Engineering Group, Department of Histology, Faculty of Medicine, Universidad de Granada, 18016 Granada, Spain; 2Instituto de Investigación Biosanitaria ibs.GRANADA, 18012 Granada, Spain; 3Doctoral Program in Biomedicine, Doctoral School, Universidad de Granada, 18016 Granada, Spain; 4Facultad de Odontología, Universidad Michoacana de San Nicolás de Hidalgo (UMSNH), Morelia 58010, Mexico; 5Department of Cell Biology, Faculty of Sciences, Universidad de Granada, 18071 Granada, Spain; 6Department of Women and Children’s Health, University of Padova, 35129 Padova, Italy; 7Corso Stati Uniti 4, Institute of Pediatric Research Città della Speranza, 35127 Padova, Italy; 8Centro de Investigaciones Biomédicas, Escuela de Medicina, Facultad de Medicina, Universidad de Valparaíso, Valparaíso 2520000, Chile

**Keywords:** umbilical cord (UC), wharton jelly stem cells (WJSC), extracellular matrix (ECM), basement membrane (BM), basal lamina (BL) stem cell-based microtissues (MT), tissue engineering (TE)

## Abstract

Wharton’s jelly stem cells (WJSC) from the human umbilical cord (UC) are one of the most promising mesenchymal stem cells (MSC) in tissue engineering (TE) and advanced therapies. The cell niche is a key element for both, MSC and fully differentiated tissues, to preserve their unique features. The basement membrane (BM) is an essential structure during embryonic development and in adult tissues. Epithelial BMs are well-known, but similar structures are present in other histological structures, such as in peripheral nerve fibers, myocytes or chondrocytes. Previous studies suggest the expression of some BM molecules within the Wharton’s Jelly (WJ) of UC, but the distribution pattern and full expression profile of these molecules have not been yet elucidated. In this sense, the aim of this histological study was to evaluate the expression of main BM molecules within the WJ, cultured WJSC and during WJSC microtissue (WJSC-MT) formation process. Results confirmed the presence of a pericellular matrix composed by the main BM molecules—collagens (IV, VII), HSPG2, agrin, laminin and nidogen—around the WJSC within UC. Additionally, ex vivo studies demonstrated the synthesis of these BM molecules, except agrin, especially during WJSC-MT formation process. The WJSC capability to synthesize main BM molecules could offer new alternatives for the generation of biomimetic-engineered substitutes where these molecules are particularly needed.

## 1. Introduction

The umbilical cord (UC) is an elongated anatomical structure with 50–60 cm length, 1–2 cm in diameter and 40 helical turns that connects the fetus to the placenta. It is developed between the fourth and seventh week of the embryonic period, lengthening to its final dimensions during the second trimester of pregnancy [1]. Histologically, the UC is basically composed of two arteries and one vein which are surrounded by an abundant amount of mucous connective tissue, the Wharton’s Jelly (WJ), and covered by amniotic epithelium. Wharton’s jelly is often divided into different connective tissue regions–the sub-amnion (SAM) region, the WJ, the intervascular (IV) and the perivascular (PV)—with all of them being scientifically relevant due to their abundant amount of mesenchymal stem cells (MSC) [2].

Nowadays, stem cells (SC) have become a promising alternative in advanced therapies and tissue engineering (TE). These undifferentiated cells are mainly characterized by their self-renewal capability and, under certain circumstances, they may differentiate into diverse lineages. Adult SC, which are usually found quiescent in different adult tissues and organs, are responsible for physiological cell renewal and also play key roles during tissue repair and regeneration processes [3]. There are several SC sources in embryonic and adult tissues, which are classified according to their differentiation potential and origin [3]. Among the SC, the MSCs are one of the most frequently used in TE research [4,5]. In this context, due to their easy harvesting, high amount of MSC, immunomodulatory properties and key cell functional features (high proliferation rate and wide differentiation capability) the Wharton’s Jelly mesenchymal stem cells (WJSC) have become a promising alternative for many TE applications, including corneal [6] or neural engineering [2,7].

The maintenance of the key features of the SCs is directly linked to their specific surrounding micro-environment, the so-called SC niche [3]. Multiple factors are related to the cell niche development and maintenance [8], such as the three-dimensional (3D) structure, the extracellular matrix (ECM) composition, the physico-chemical microenvironmental conditions (pH, growth factors and metabolite concentration for instance) [9] and the cell–cell or cell–ECM interactions [8,9,10]. As there are different types of tissues and organs, there are also multiple SC niches with specific microenvironmental features [10]. Some niches may consist of adjacent cells in different functional or maturation states; different ECM molecules, such as the basement membrane (BM) in the epithelial tissues; and some basal membrane-like macromolecular structures, such as in muscle tissue [11,12].

The BM are dynamic structures with different ultrastructural patterns, molecular composition, dimensions and functions among tissues [11,13]. These are essential structures to attach epithelial cells to the subjacent connective tissue [11], but also can act as a molecular selective barrier, growth factor reservoir, intracellular signaling pathways, cell differentiation and proliferation, tissue repair [11] or even as an efficient mechano-transductor [14]. At transmission electron microscopy (TEM), the BM may range from 50 to 100 nm of thickness, being ultra-structurally divided into two layers, the basal lamina (BL) and the lamina fibro-reticularis. The BL is composed of two well-defined structures, the lamina lucida (or rara) and the lamina densa [11,15]. The four main ECM components of most BM are collagen type IV (Col IV), the glycoproteins laminin (LAM) and nidogen/entactin-1 (NID-1) and heparan sulfate proteoglycan-2 or perlecan (HSPG2) [11,13]. However, other ECM molecules can be found within some BM, such as the proteoglycan agrin (AGRN) or the collagen type VII (Col VII), with distinctive functions. Curiously, in some tissues, the lamina fibro-reticularis is absent, and a BL or a BL-like pericellular matrix remains delimiting or surrounding the cells. Well-known non-epithelial cell types surrounded by these complex ECM structures are the myofibers, Schwann cells, perineural cells, chondrocytes or adipocytes. This molecularly complex pericellular matrix provides developmental cues and plays distinctive structural, functional and regenerative roles [3,16]. 

Despite the widespread use of WJSC in SC research, the structural features of their cell niche are still poorly understood. Few early studies showed the presence of some BM-related ECM molecules within UC [17,18], but most of the current literature is mainly focused on the potential usefulness of these cells in TE [19,20,21,22]. In this regard, recent ex vivo studies demonstrated that the WJSC can produce an important amount of ECM under culture conditions. These cells grew fast and produced ECM-rich cell sheets in two-dimensional (2D) standard cell-culture conditions [23]. Interestingly, by using the scaffold-free microtissues (MT) technique, it was demonstrated that WJSC generated stable, viable and ECM-rich MT for a wide range of applications [24,25]. Indeed, WJSC-derived MTs (WJSC-MTs) progressively deposited different ECM molecules, including Col IV, which acquired a well-defined pattern over time ex vivo [24]. These studies support the hypothesis that the WJSC functional profile could be influenced by their own ECM niche, including BM-related molecules, which were even required during different ex vivo culture-condition techniques. In this context, the aim of the present study was to evaluate the presence of the main BM-related molecules that could be part of the WJSC niche in full-term UC as well as under 2D standard culture conditions and during the complex ex vivo MT formation process. 

## 2. Materials and Methods

### 2.1. UC Tissue Samples

In this study, six full-term UC segments (approximately 4–5 cm length) from six independent births were obtained under signed consent. Fresh UC segments were collected in cold transport media (DMEM + 2% antibiotics/antimycotics (both from Merck, Darmstadt, Germany)) and samples for histology (light and transmission electron microscopy (TEM)) and cell culture were processed on the same day as follows (Figure 1).

### 2.2. Histological Analyses

The UC tissue samples were transversally segmented into 0.3–0.5 cm thickness sections, washed in 0.1M phosphate saline buffer (Merck, Darmstadt, Germany) and fixed in 3.7% neutral buffered formaldehyde (Panreac Química, Barcelona, Spain) for at least 48 h at room temperature (RT). Then, samples were dehydrated in increasing ethanol solutions, cleared in xylol and embedded in paraffin following established protocols [26,27,28]. Histological sections of 5-µm thick were subjected to histological, histochemical and immunohistochemical staining for the analysis of the four different regions that compound the UC (Figure 1).

The main histological features of the UCs, the 2D cell cultures and the WJSC-MTs (see tissue processing below) were analyzed by hematoxylin–eosin staining (HE). Moreover, a screening of the ECM collagen fibers, proteoglycans and glycoproteins was performed by using Picrosirius staining (PS), Alcian blue (AB) pH 2.5 and Periodic Acid–Schiff (PAS) histochemical methods, respectively. To determine the presence of BM-specific ECM molecules, indirect immunohistochemical procedures for collagens (type IV (Col IV) and type VII (Col VII)), proteoglycans (heparan sulfate proteoglycan 2 (HSPG2) and agrin (AGRN)) and glycoproteins (laminin (LAM) and nidogen/entactin-1(NID-1)) were performed. Technical information related to the immunohistochemical procedures used (antibodies, pretreatments and references) are summarized in Table 1. Histochemical and immunohistochemical procedures were performed following previously described methods [5,19,29,30,31]. In immunohistochemistry protocols, the incubation of the primary antibody was omitted in independent samples as a negative technical control (results available in Appendix A). Moreover, the amnios and UC-blood vessels served as internal positive control for histochemical and immunohistochemical reactions. 

### 2.3. WJSC Isolation and Culture

WJSC were isolated from healthy full-term UCs biopsies as detailed in previously described protocols [19,24]. Briefly, amnios and blood vessels were removed from UCs. The WJ region was isolated and digested in collagenase type I solution (Gibco BRL, Grand Island, NY, USA; Cat. nº: 17018-029) and cells were then collected by centrifugation (1000 rpm during 10 min). Cells were cultured with Amniomax™ culture media (Gibco BRL, Grand Island, NY, USA; Cat. nº: 17001-074) with 1% antibiotics/antimycotics cocktail solution (Merck, Darmstadt, Germany; Cat. nº: A-5955). WJSC were kept under standard culture conditions (37 °C and 5% CO_2_) until sub-confluence. Later, WJSC were detached with trypsin–EDTA (Merck, Darmstadt, Germany; Cat. nº: T3924) and subcultured until the fourth/fifth passage to obtain sufficient cells for this study. Furthermore, the WJSC MSC profile was confirmed by flow cytometry using a NovoCyte Flow Cytometer (ACEA Biosciences, San Diego, CA, USA) against a panel of positive (CD73 (98.72%), CD90 (98.15%), CD105 (95.63%)) and negative (CD 45, CD34, CD11b, CD19 and HLA-DR) markers (BD Biosciences, Franklin Lakes, NJ, USA; Cat. nº: 562245) as previously described [24,32].

### 2.4. Agarose Microchips and MT Generation

To generate the WJSC-MTs, agarose microchips technique was applied following a well-described procedure [24,33]. Briefly, to elaborate agarose microchips, a 3.5% agarose solution was prepared diluting type I agarose in phosphate buffered saline buffer (PBS) (both from Merck, Darmstadt, Germany; Cat. nº: A-4718 and D8662) and subsequently autoclaved. Agarose solution was melted and poured into commercially available polydimethylsiloxane (PDMS) molds (Merck, Darmstadt, Germany) which contained 256 microwells with 400 µm of diameter and 800 µm of depth each. PDMS molds containing agarose solution were placed at RT until gelation allowing to obtain agarose microchips with a negative replica of the PDMS molds used. Then, agarose microchips were carefully harvested and stored in PBS at 4 °C until their use. To generate the WJSC-MTs, the agarose microchips were first equilibrated at 37 °C during 30 min with 2 mL of Amniomax™ culture media in 12 well culture plates. The culture medium was removed and 5 × 10^4^ WJSC, in 190 µL of culture medium, was seeded in each agarose microchip. Plates were placed at 37 °C for 1 h to allow the cells to settle into the wells by gravity and 1 mL of culture medium was added around the microchips. The WJSC-MTs were kept under standard culture conditions (37 °C at 5% CO_2_) and harvested for evaluation at 4, 7, 14, 21 and 28 days of ex vivo development (EVD). 

### 2.5. Ex Vivo 2D Cell Cultures and WJSC Derived MTs Histological Analyses

Histological and histochemical analyses of WJSC 2D cultures and WJCS-MTs were assessed on three independent samples. In the case of the WJSC 2D cultures, 1 × 10^4^ cells were seeded per chamber by using commercially available cell culture slides (Thermo Fisher Scientific, Waltham, MA, USA; Cat. nº: 154526). Cells were kept in standard culture conditions until confluence, washed in PBS, chemically fixed (3.7% neutral buffered formaldehyde), air-dried and stored at −20 °C until use. In order to conduct histological analyses of the WJSC-MTs, they were processed as a tissue. Briefly, agarose chips containing MTs were carefully harvested, chemically fixed and processed with an adapted tissue processing (dehydration, clearing and paraffin-embedding) technique which consisted in the use of slight centrifugation steps between each change as described previously [24,33]. Then, 5-µm thick sections of WJSC-MTs and chamber slides containing WJSC were stained with the same histochemical and immunohistochemical panel described above (see Section 2.2).

### 2.6. Ultrastructural Analyses

For TEM, UCs samples and WJSC-MTs at 4 and 7 EVD were fixed in 2.5% buffered glutaraldehyde (Panreac Química, Barcelona, Spain; Cat. nº: 163857.1611) in 0.05 M cacodylate buffer pH 7.2 (Merck, Darmstadt, Germany; Cat. nº: C-0250) at 4 °C for 4 h and then washed three times in the same buffer at 4 °C and postfixed with 1% osmium tetroxide (Electron Microscopy Sciences, Hatfield, PA, USA; Cat. nº: 19110). Samples were dehydrated and finally embedded in EMBed-812 resin embedding kit (Electron Microscopy Sciences, Hatfield, PA, USA; Cat. nº: 14120) by using standard procedures. Semithin sections (1-µm thickness) were conducted and stained with toluidine blue (Merck, Darmstadt, Germany; Cat. nº: T-3260). Once the area was selected, ultrathin sections (50 nm) were made and stained with uranyl acetate (Merck, Darmstadt, Germany; Cat. nº: 8473). Ultrathin sections were analyzed with a transmission electron microscope Libra 120 Plus (Carl Zeiss SMT, Oberkochen, Germany).

### 2.7. Statistical Analyses

The percentage of positive cells and the mean of staining intensity for BM component (Col IV, Col VII, HSPG2, AGRN, LAM and NID-1) detected by immunohistochemistry were quantified in six independent UCs samples using ImageJ software (National Institutes of Health, Bethesda, MD, USA) as previously described [19,34]. Results were subjected to the Shapiro–Wilk normality test and statistical differences (*p* < 0.05) among SAM, WJ, IV and PV groups were determined by Mann–Whitney non-parametric test using RealStatistics software (University, West Lafayette, IN, USA).

## 3. Results

### 3.1. Histological, Histochemical and Immunohistochemical Results of Full-Term UCs

Histological evaluation of transversally sectioned UC samples by HE ascertained the presence of an amniotic stratified epithelium covering the entire length of the cord, an underlying gelatinous connective tissue, rich in WJSC, and three blood vessels (two arteries and one vein) immersed in it. Regarding the UC stromal tissue, the four different regions were analyzed. In the SAM region, cells were immersed in a less dense ECM and they were found with a circular or longitudinal orientation to the long axis of the organ. In the WJ region, the cells were elongated and formed lineal and interconnected groups with a clear circular orientation. Within the IV region, the cells were abundant, homogeneously distributed and preferably oriented parallel to the long axis of the UC. Finally, in the PV region, cells were found with an elongated and rounded morphology indicating a more spiral distribution around the vascular blood vessels walls (Figure 2).

Different histochemical methods were performed in UCs to identify the presence of main ECM components. For proteoglycan and glycoproteins’ content, two histochemical methods were applied. First, the AB method showed a high amount of acid proteoglycans ubiquitously distributed in the ECM of different UC regions, being slightly less positive in PV zone. Moreover, PAS histochemical method revealed PAS-positive pericellular reaction as well as some cells with PAS positive cytoplasmic granules (Figure 2). Regarding the collagen network, it was assessed by PS staining revealing a homogeneous distribution of relatively thin collagen bundles throughout the ECM of the UC. Collagen bundles were well-defined within the WJ and IV regions while they were less dense and irregularly distributed within the SAM region (Figure 2). Please note that the histochemical methods used to characterize main ECM molecules, such as AB, PAS and PS, has been used within previous works [19,35].

The presence of specific BM components was analyzed by indirect immunohistochemistry, showing a clear positive staining in amniotic BM and WJSC pericellular matrix for all molecules and in all regions analyzed (Figure 3). Analysis of Col IV, a major component of the BM, showed a consistent positive reaction in the amniotic BM and WJSC pericellular matrix. Interestingly, a significant superior percentage of positive cells and intensity for Col IV was observed in the SAM region (*p* < 0.05) (Figure 3 and Figure 4). In contrast, analysis of Col VII, another collagen found in some specific BM, displayed a higher percentage of positive cells and intensity in SAM and WJ regions than other zones (*p* < 0.05). Analyzing the expression of the proteoglycans (HSPG2 and AGRN) and glycoproteins (LAM and NID-1), clear positivity was observed. Statistical analyses revealed significant differences in percentage of positive cells and intensity of these immunohistochemical reactions in SAM and WJ regions as compared to IV and PV zones (see details in Figure 4). Surprisingly, the expression of NID-1 was not just restricted to amniotic BM and WJSC pericellular matrix, actually NID-1 was also slightly positive in UC stroma (Figure 3 and Figure 4).

### 3.2. Ex Vivo 2D Cell Cultures and WJSC Derived MTs Histological Results

Histological and histochemical analyses of WJSC-2D and WJSC-MT showed clear differences between them. Concerning the WJSC-2D cell cultures, HE displayed the typical elongated morphology of these MSCs. Histochemistry showed a slightly positive reaction for AB histochemical method as well as for PAS staining confirming the synthesis of proteoglycans and glycoproteins by these cells. However, no well-formed collagen fibers were observed within these cell cultures by PS histochemical methods (Figure 5). 

The analysis of the WJSC-MT revealed the formation of a dynamic 3D spherical structure during the study. Initially, WJSC-MTs presented a homogeneous cell distribution in the MT. However, a maturation process was observed from day 7 onwards, when cells progressively acquired a peripheral pattern and produced an ECM-rich inner core (ECM-core). The evaluation of the proteoglycans by AB showed a clear extracellular synthesis of these molecules by the WJSC-MT over the time and in function of the cell distribution. Moreover, the PAS histochemical method revealed the synthesis of glycoproteins within the WJSC-MT. This reaction was intercellular between 4 and 7 EVD and acquired a membranous pattern, around the ECM-core between 14 and 28 EVD (Figure 5). Regarding PS staining, it was slightly positive over the time between the cells and within the ECM-core (Figure 5). 

Immunohistochemical analyses for BM components showed positive expression in both cell culture conditions for BM collagens, proteoglycans and glycoproteins but clear differences were found in positivity and 3D distribution between WJSC-2D and WJSC-MT.

Immunohistochemical analyses of WJSC-2D revealed the slight synthesis capacity of major BM molecules, such as Col IV and the glycoprotein NID-1 with slightly higher expression of HSPG2 and the glycoprotein LAM. In contrast, no positive immunohistochemical reaction for collagen Col VII and proteoglycan AGRN were found (Figure 6). 

Attending to the BM molecular profile of WJSC-MT, a clear maturation process occurred over the time. Initially, a pericellular expression for Col IV was found but from 7 EVD onward, a clear ECM-core positivity was observed progressing to a distinctive peripheral membranous pattern, delineating the cell ECM-core interface, especially noticeable at 28 EVD (Figure 6). Col VII displayed a similar pattern than Col IV, but its positive reaction was stronger and more homogeneous within the ECM-core during the time (Figure 6). The expression of proteoglycans HSPG2 and AGRN showed clear differences. Whilst HSPG2 was consistently positive at both cellular and ECM-core levels throughout the study, AGRN was entirely negative (Figure 6). Finally, the analysis of LAM and NID-1 glycoproteins also showed clear differences. For LAM, a weak pericellular reaction was observed at 4 EVD, being expressed within the ECM-core from 7 EVD onwards and adopting a homogeneous pattern that progressed to well-recognized positive zones between 21 and 28 EVD (Figure 6). In contrast, a homogeneous distribution of NID-1 was observed within the WJSC-MT over the time without a clear pericellular or ECM-core pattern as noticed with other molecules. (Figure 6).

### 3.3. Ultrastructural Analyses of Full-Term UCs and WJSC-MT by Transmission Electron Microscopy

Finally, an ultrastructural analysis of full-term UCs samples and WJSC-MTs was conducted by TEM. The analysis of the UCs revealed that the WJSCs are immersed in an ECM composed of collagen fibrils and a high amount of non-fibrillar ECM or growth substance. The collagen fibrils, with their characteristic cross bands, were loosely grouped and surrounded by a poorly organized fibrous material (Figure 7). The analysis of the WJSC within the UCs showed that each cell was covered by a discontinuous and thin pericellular matrix composed of a lamina lucida and a thin lamina densa resembling the typical structure of the BL. Hemidesmosomes or focal contact were not observed by TEM analysis. In addition, full-term WJSC contain few organelles (mitochondria, RER and vesicles) and a variable number of small accumulations compatible with glycogen granules which support the PAS positive reaction described above (Figure 7). WJSC-MT ultrastructural analyses showed a poorly structured BL-like pericellular matrix around some cells (Figure 7). Concerning the ECM-core, TEM showed an apparently poorly organized fibrillar materials without the presence of mature collagen fibrils (Figure 7). Moreover, WJSC at the MT surface showed an active release of numerous extracellular vesicles (Figure 7).

## 4. Discussion

The BM is an essential structure for the proper function of various tissues, and often it forms part of some SC niches [3,10,11]. In this study, a comprehensive histological analysis of the main BM molecules, that could be part of the WJSC niche, was conducted in full-term UC. In addition, the expression profile of these molecules was further histologically analyzed in WJSC 2D culture technique and during WJCS-MT formation ex vivo.

In this study, histology revealed differences in the cellular and ECM composition among UC regions. Interestingly, the distribution and spatial orientation of the WSJC and surrounding ECM varied among the UC zones between circular, parallel or even spiral to the long axis of this organ. These findings may indicate that each UC zone is characterized by a distinctive histological pattern, being consistent with previous descriptions [2,19,35]. Accompanying this distinctive histological pattern, the histochemistry of main ECM molecules revealed a high content of proteoglycans (AB staining) and collagen fibers (PS staining) within each UC zone, which, as previously reported, are key features of this highly specific mucous connective tissue [19,35]. In addition, PAS staining suggests the presence of a thin pericellular matrix around the WJSC as well as glycogen granules. The PAS-positive pericellular matrix may indicate the presence of BM-related glycoproteins. Concerning the glycogen granules, confirmed by TEM analysis, they indicate that WJSC store glycogen for their metabolic activities as these cells are immersed in a highly abundant, hydrated and uncapillarized mucous ECM. These features are not specific of WJSC, since other mesodermal cells (such as chondrocytes or muscle fibers) also store glycogen or even lipids [36]. The UC is formed early during development, and from then it is constantly subjected to strain, torsion or even compression forces. Therefore, the cell morphology, metabolic requirements, and ECM molecular composition and pattern of the UC could be related with these constant structural and physical needs [37,38].

Furthermore, as observed by the histochemical evaluation of the WJSC within full-term UC, these cells could be surrounded by a thin PAS-positive pericellular matrix. A classical study revealed the presence of some BM-related molecules (Col IV, LAM, HSPG and Col VII) and other ECM within the UC stroma [17]. However, a comprehensive analysis of the distribution pattern and especially their relationship with the WJSC was not described. More recent histological studies confirmed the presence of collagen type IV in UC stroma [19] and a clear membranous structure was obtained during WJSC-MT formation ex vivo [24]. In this context, our study delved into the complete analysis of the ECM around the WJSCs in order to demonstrate the presence of a BL-like pericellular matrix which form part of the WJSC niche. Regarding the fibrillar components that form part of most BM, our study confirms the presence of collagens type IV and VII. These structural ECM fibers showed a clear pericellular pattern around the WJSC within all UC zones. Furthermore, immunohistochemistry of non-fibrillar ECM molecules demonstrated a consistent expression of BM-related proteoglycans (HSPG2 and agrin) and glycoproteins (laminin and nidogen-1) surrounding each WJSC. The presence of all these BM-related molecules clearly supports the fact that WJSC within UC are surrounded by a thin and molecularly complex BL-like pericellular matrix such as occurred with other well-known non-epithelial cell types [36,39]. Despite the clear BM molecular profile observed around the WJSC, in this study TEM analyses were conducted to determine if these cells are or not surrounded by a BM or BL-like pericellular matrix. These analyses confirmed the presence of a thin and discontinuous pericellular matrix composed of a lamina lucida and a thin lamina densa. Therefore, immunohistochemical and TEM analyses support the hypothesis that the WJSCs, within UC, build and maintain a pericellular matrix that molecularly and ultrastructurally resemble a BL. 

The presence of this BL-like structure could be related to structural and biomechanical requirements within the UC. The WJ, the most abundant element of the UC stroma, is composed of a collagen network surrounded by abundant hydrophilic non-fibrillar ECM molecules produced by the WJSC, which represent the main physiological function of these cells in this organ. The main role of the WJ within the UC is still poorly understood, but a few recent studies attribute a key biomechanical role to it [37,40]. Indeed, using equibiaxial tension test and computational approaches it was demonstrated that the WJ stiffness plays an important protective role avoiding the biomechanical failure of the UC blood vessels (compression, torsion, collapse, etc.) ensuring a continuous materno–fetal blood supply [37]. In this context, the presence of a BL-like pericellular matrix could be closely related to the WJ biomechanics. Therefore, the BL-like pericellular matrix described here—through the network formed by laminin, HSPG2, agrin, nidogen-1 and collagens type IV and VII—could provide a physical substrate to integrate the WJSC within the UC stroma keeping the well-defined histological pattern and cell distribution in each UC zone [13,41]. In addition, the presence of HSPG2 could provide to this BL-like pericellular matrix mechano-transductor properties which could provide information to the WJSC about the UC dynamics to regulate ECM turnover [41,42]. Finally, it is well-known that glycoproteins and specifically proteoglycans present a wide range of growth factor-binding sites [41,42,43]. It is probable that the BL-like pericellular matrix described here could also serve as a growth factor reservoir. However, future studies are needed to determine the exact profile of growth factors present within the WJSC niche under physiological conditions. 

The present study also evaluated histologically the capability of the WJSC to synthesize these main BM molecules under 2D and 3D cell cultures conditions. Our histochemical results demonstrated that the MT-forming WJSC synthetize an abundant ECM, which was not obtained with standard 2D cell culture techniques. These results are in concordance with previous works where the use of 3D cell culture techniques demonstrated to be an efficient manner to generate natural ECM ex vivo [5,8]. Moreover, the high amount of collagens, proteoglycans and glycoproteins synthesized by WJSC-MTs in this work is well-supported by a previous study conducted with these particular cell lineage [24]. Interestingly, WJSC-MT showed a clear and progressive expression of most BM-related molecules studied here. Histochemistry confirmed the synthesis of an ECM-core and how the WJSC progressively acquired a superficial distribution over the time within the WJSC-MT. PAS and collagen type IV staining suggested the formation of a membrane-like structure delineating the interphase between cells and the ECM-core generated. However, the other BM-related molecules studied—collagen type VII, laminin, nidogen-1 and HSPG2—although clearly positive, did not develop a membranous pattern, and instead showed a more homogeneous distribution within the formed MT. Surprisingly, agrin was not produced by the WJSC in 2D or 3D cell cultures, probably due to a lack of stimuli and maturation. In this line, TEM analysis revealed a thin but not well-structured or immature BL-like pericellular matrix around some MT-forming WJSC ex vivo. Therefore, these results demonstrate that the WJSC can synthesize most of the main BM ECM molecules ex vivo, especially when MT technique is used. These molecules did not acquire the complex organization and pericellular pattern observed within full-term UC, but they were clearly positive at the extracellular level, except for agrin. The complex process of self-assembly of BM is initiated by the cell–laminin interaction followed by the binding and polymerization of collagen type IV network. This structure is further stabilized by the NID-1 and HSPG2 [13]. However, it was also reported that the self-assembly ability collagen type IV [13] explained the membranous pattern acquired by this molecule within the WJSC-MT. In view of our results, we hypothesize that the expression of BM-related ECM molecules by the WJSC-MT may play supportive roles for cell growth, migration and the 3D pattern acquired by these cells during the MT forming process. Therefore, the differences observed between UC and cell cultures could be explained due to the lack of biological and physical factors in the 2D and 3D culture conditions. Actually, none of these cell culture methods were able to recreate the complex processes involved during UC development as expected, these results being supported by the partial differentiation observed within other models such as bioengineered skin [44,45] or cartilage substitutes [33,46,47]. 

The capability of the WJSC to synthesize BM-related molecules ex vivo, especially during MT formation, could represent a new alternative for diverse TE applications where BM ECM molecules are particularly needed. For example, it is well-accepted that the molecular composition and thickness of the BL of peripheral nerve fibers is important in nerve tissue regeneration [29]. Actually, laminin is considered a key factor in this field as it efficiently promotes Schwann cell proliferation, migration and bands of Büngner formation acting as efficient molecular support for axonal regrowth [26,29,48,49]. Another example can be found in cartilage, where chondrocytes are surrounded by a specific pericellular matrix—composed by collagen type IV, laminin, HSPG2 and nidogen-1 among other ECM molecules—that plays well-defined protective, structural and regenerative roles [50,51,52,53,54]. Therefore, the well-defined capability of the WJSCs to synthesize fibrillar and non-fibrillar main BM molecules, specifically by using the MT technique, could be used to generate more functional and biomimetic-engineered models for skin, mucosal structures, peripheral nerve or cartilage TE applications. Moreover, our study demonstrated a consistent expression of highly specific BM molecules by these cells (within the UC and ex vivo) and thus they could be used as new distinctive markers to define the WJSC phenotype in tissue engineering protocols.

In summary, our results demonstrate the expression of the main BM molecules, with a well-defined pericellular distribution pattern, around the WJSCs in each UC zone. This highly specific and complex pericellular matrix resembles the molecular composition and ultrastructural features of the BL. This BL-like pericellular matrix could play essential roles within the WJSC niche, providing supportive, integrative, molecular or even mechano-transduction properties. Moreover, ex vivo analyses demonstrate the synthesis capability of the WJ region-derived WJSC to generate enhanced cellular and BM molecules-rich MTs. These WJSC-MTs represent new alternatives for many TE applications, with special interest in tissues where key BM components are specially needed. Overall results obtained in this work support the hypothesis that a BL-like pericellular matrix is essential for WJSC forming a well-defined niche within the UC stroma. In addition, these BM-specific molecules could serve as a new panel of markers to define the WJSC phenotype in TE. Finally, future studies are still needed to determine the biological significance and regenerative potential of the BM molecules produced by the WJSCs in diverse TE applications. 

## Figures and Tables

**Figure 1 cells-12-00629-f001:**
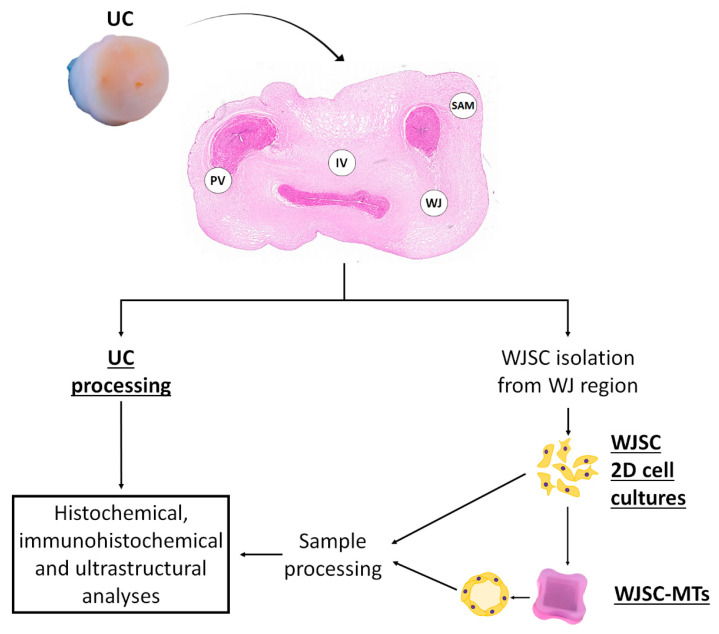
Flow chart of the experimental procedures applied to umbilical cord in this study. A representative image of an agarose microchip is shown at the bottom of the image. UC: umbilical cord; SAM: sub-amnion; WJ: Wharton’s jelly; IV: intervascular and PV: perivascular region; WJSC-MTs: Wharton’s Jelly stem cell-derived microtissues.

**Figure 2 cells-12-00629-f002:**
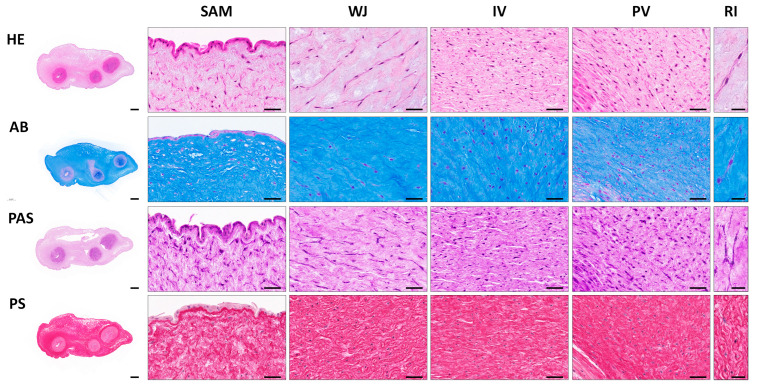
Histochemical analyses of full-term human UCs. Hematoxylin–eosin (HE), Alcian Blue (AB), Periodic Acid-Schiff (PAS) and Picrosirius Red (PS) methods were applied to analyze general morphology and synthesis of specific ECM components in complete UC (left) and specific regions; Sub-amnion (SAM); Wharton’s Jelly (WJ); intervascular (IV) and perivascular (PV). Representative images (RI) at higher magnification are shown on the right. Scale bar: 1000 µm (complete UC), 50 µm (specific UCs regions) and 25 µm (RI).

**Figure 3 cells-12-00629-f003:**
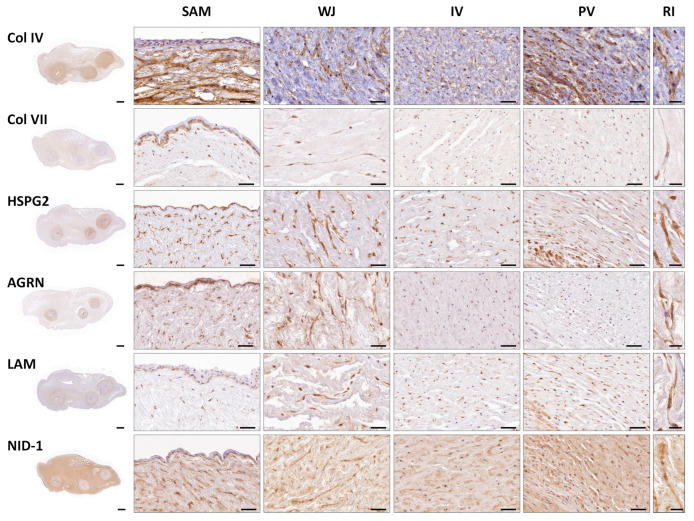
Immunohistochemical analyses of the basement membrane (BM) molecules (collagen type IV and VII (Col IV) (Col VII), proteoglycans heparan sulfate proteoglycan 2 (HSPG2) and agrin (AGRN) and glycoproteins laminin (LAM) and nidogen-1 (NID-1)) in full-term human UCs. Immunohistochemical methods were applied to analyze complete UC (left) and specific regions; Sub-amnion (SAM); Wharton’s Jelly (WJ); intervascular (IV) and perivascular (PV). Representative images (RI) at higher magnification are shown on the right. Brown color: immunohistochemical positive reaction and slight blue: hematoxylin contrast. Scale bar: 1000 µm (complete UC), 50 µm (specific UC regions) and 25 µm (RI).

**Figure 4 cells-12-00629-f004:**
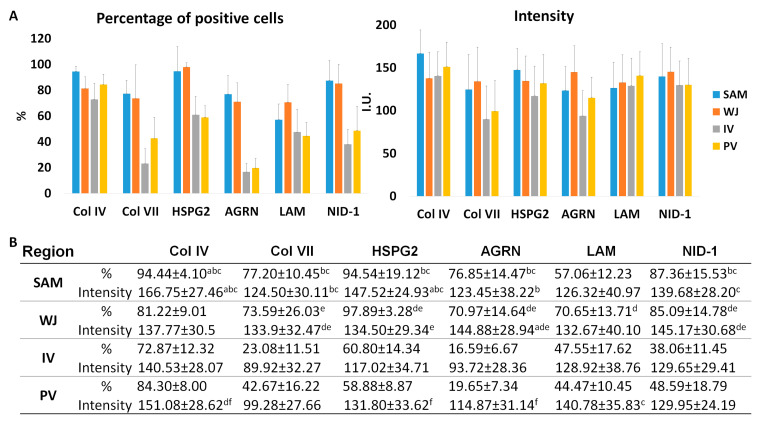
Graphical representation (**A**) and quantitative data (**B**) of percentage of positive cells and positive signal intensity (I.U: intensity units) for BM components (collagen type IV and VII (Col IV and Col VII), proteoglycans heparan sulfate proteoglycan 2 (HSPG2) and agrin (AGRN) and glycoproteins laminin (LAM) and nidogen-1 (NID-1)) expression by regions in full-term human UCs. Results are shown as mean ± SD values for each variable. Sub-amnion (SAM), Wharton’s Jelly (WJ), intervascular (IV) and perivascular (PV). Significant differences between regions are indicated as follows; a: SAM vs. WJ; b: SAM vs. IV; c SAM vs. PV; d: WJ vs. PV; e: WJ vs. IV; f: IV vs. PV.

**Figure 5 cells-12-00629-f005:**
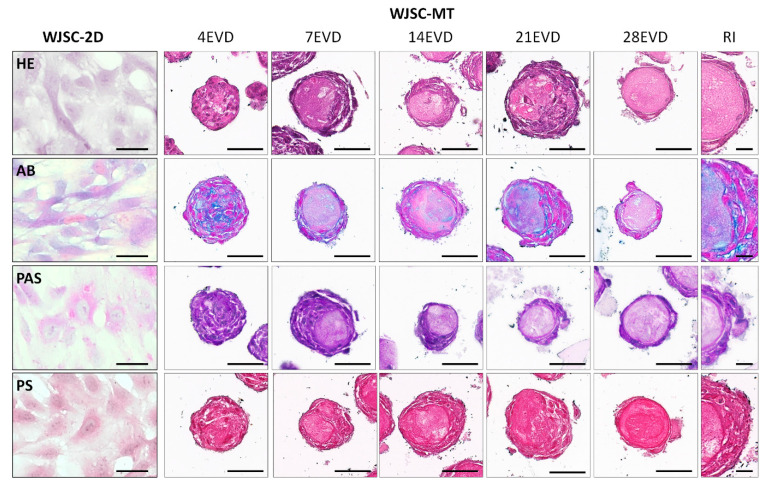
Histochemical analyses of WJSC cultured under two-dimensional (WJSC-2D) and three-dimensional microtissue (WJSC-MT) culture methods at different days of ex vivo development (EVD). Hematoxylin–eosin (HE), Alcian Blue (AB), Periodic Acid-Schiff (PAS) and Picrosirius Red (PS) methods were applied to analyze general morphology and synthesis of specific ECM components in WJSC-2D and WJSC-MT at 4, 7, 14, 21 and 28 days of EVD. Representative images (RI) at higher magnification are shown on the right. Scale bar: 50 µm and 25 µm (RI).

**Figure 6 cells-12-00629-f006:**
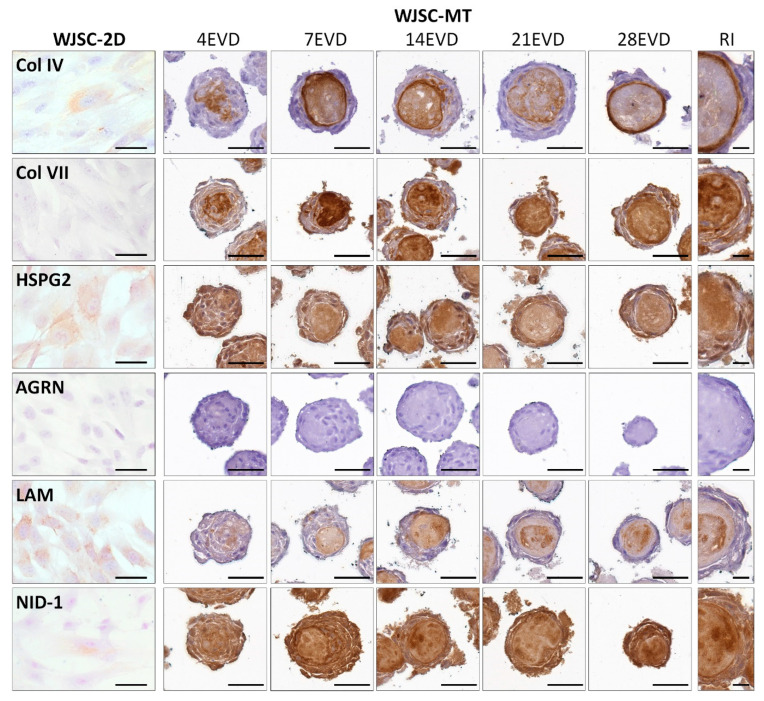
Immunohistochemical analyses of the basement membrane (BM) molecules (collagens type IV and VII (Col IV) (Col VII), proteoglycans heparan sulfate proteoglycan 2 (HSPG2) and agrin (AGRN) and glycoproteins laminin (LAM) and nidogen-1 (NID-1)) expressed in WJSC-2D and WJSC-MT culture methods at 4, 7, 14, 21 and 28 days of EVD. Representative images (RI) at higher magnification are shown on the right. Scale bar: 50 µm and 25 µm (RI).

**Figure 7 cells-12-00629-f007:**
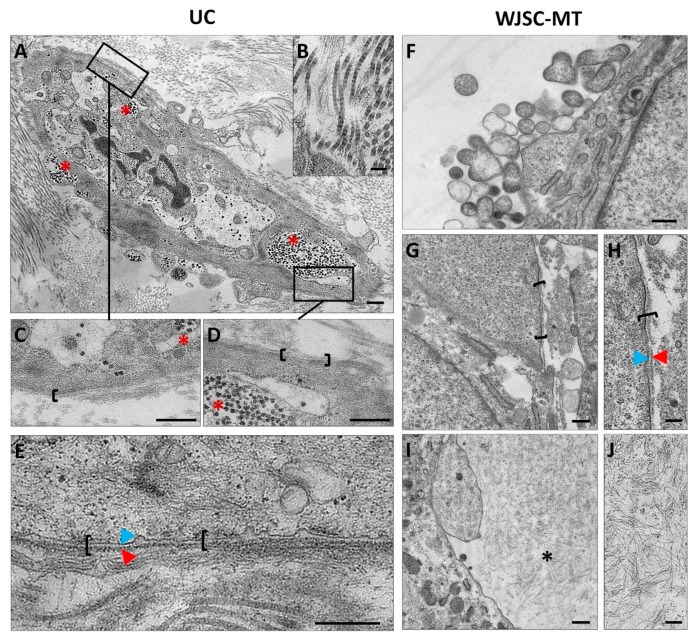
Ultrastructural analyses by transmission electron microscopy (TEM) of Wharton’s Jelly-derived stem cells (WJSC) in full-term umbilical cord (UC) and WJSC-derived microtissues (WJSC-MT). Large amounts of intracellular glycogen (red asterisks) are seen in a lower magnification cell image (**A**) and magnified areas (**C**,**D**). A representative image of collagen fibers and non-fibrillar components of ECM full-term UC is represented in (**B**). A discontinuous basal lamina-like (BL) pericellular structure (black brackets), formed by lamina lucida (blue arrowhead) and lamina densa (red arrowhead), is also visible in full-term UC (**C**–**E**) and 4 days of ex vivo development (EVD) WJSC-MT (**G**,**H**). Large amount of extracellular vesicles in WJSC-MT at 4EVD (**F**) and non-fibrillary content in extracellular matrix core (black asterisk) was observed in WJSC-MT at 7EVD (**I**,**J**). Scale bar: 500 nm (**A**,**C**–**G**,**I**) and 200 nm (**B**,**H**,**J**).

**Table 1 cells-12-00629-t001:** Antibodies and technical conditions for immunohistochemical analyses. All primary antibodies were diluted in PBS 0.1M (pH 7.2–7.4) with 0.3% Tween-20 (Merck, Darmstadt, Germany. Cat. nº: P1379).

Antibody	Dilution/Incubation	Pretreatment	Reference
Mouse monoclonal anti-collagen type IV	Prediluted Overnight at 4 °C	EDTA buffer pH 8 20 min at 95 °C	Master Diagnóstica (Granada, Spain) (Cat. nº: MAD001060QD)
Rabbit polyclonal anti-collagen type VII	1:50 Overnight at 4 °C	EDTA buffer pH 8 20 min at 95 °C	Novus Biological (Englewood, CO, USA) (Cat. nº: NBP2-37900)
Rabbit polyclonal anti-HSPG2	1:250 Overnight at 4 °C	EDTA buffer pH 8 20 min at 95 °C	Abbexa (Cambridge Science Park, UK) (Cat. nº: abx103270)
Rabbit polyclonal anti-agrin	1:500 Overnight at 4 °C	Citrate buffer pH 6 20 min at 95 °C	Abbexa (Minneapolis, MN, USA) (Cat. nº: abx037897)
Rabbit polyclonal anti-laminin	1:250 Overnight at 4 °C	Citrate buffer pH 6 20 min at 95 °C	Abcam (Bristol, UK) (Cat. nº: ab11575)
Goat polyclonal anti-nidogen/entactin-1	1:30 Overnight at 4 °C	EDTA buffer pH 8 20 min at 95 °C	R&D Systems (Minneapolis, MN, USA) (Cat. nº: AF2570)
ImmPRESS^®^ HRPAnti-Mouse IgG (Peroxidase)	1 h at RT	-	Vector Laboratories (Burlingame, CA, USA) (Cat. nº: MP-7401)
ImmPRESS^®^ HRP Anti-Rabbit IgG (Peroxidase)	1 h at RT	-	Vector Laboratories. (Burlingame, CA, USA) (Cat. nº: MP-7402)
ImmPRESS^®^ HRP Anti-Goat IgG (Peroxidase)	1 h at RT	-	Vector Laboratories. (Burlingame, CA, USA) (Cat. nº: MP-7405)

## Data Availability

The data presented in this study are available on request from the corresponding author.

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
