# Peer review of "Expression of Basement Membrane Molecules by Wharton Jelly Stem Cells (WJSC) in Full-Term Human Umbilical Cords, Cell Cultures and Microtissues"

_cells, 2023, doi:10.3390/cells12040629_

Round 1

Reviewer 1 Report

The authors provide a highly interesting and relevant study on the expression and distribution of multiple ECM components in the umbilical cord using histochemical and immunohistochemical methods. Moreover, ex vivo culturing of Wharton’s Jelly stem cells derived microtissues is performed and TEM was used to show ultrastructural details of UC and WJSC-MT. The paper is well-structured, well-written and features organized and informative graphical representation of the major findings. However, some minor spelling corrections and additional information should be provided.

(1) Please be more consistent with adjectives that have a prefix such as "non-". Use "non-" with or without a hyphen "-" consistently.

(2) Please provide at least one supplement figure with negative controls for IHC stainings for every antibody used in the study.

(3) In figure 2 and 3, please correct abbreviations in the figure itself and the supporting figure text (e.g. SA / SAM, WH ( WJ, etc.).

(4) When providing quantitative data (e.g. in Fig. 4), please indicate which values are shown (mean, median, etc. with standard deviation or standard error, or other).

(5) Please carefully read and revise or rewrite several sentences marked in yellow in the pdf document provided with this revision letter.

Author Response

The authors provide a highly interesting and relevant study on the expression and distribution of multiple ECM components in the umbilical cord using histochemical and immunohistochemical methods. Moreover, ex vivo culturing of Wharton’s Jelly stem cells derived microtissues is performed and TEM was used to show ultrastructural details of UC and WJSC-MT. The paper is well-structured, well-written and features organized and informative graphical representation of the major findings. However, some minor spelling corrections and additional information should be provided.

R/: Dear reviewer, we really appreciate the feedback and positive comments that helped us to improve our manuscript. In this sense, we introduced suggested changes in the manuscript, and we hope it will be suitable for publication in Cells.

(1) Please be more consistent with adjectives that have a prefix such as "non-". Use "non-" with or without a hyphen "-" consistently.

R/ Thank you for your helpful comments. Errors have been corrected to be more consistent throughout the manuscript.

(2) Please provide at least one supplement figure with negative controls for IHC stainings for every antibody used in the study.

R/: Thank you for your requirement, it will provide more reliable results. A figure with negative technical controls showing no IHC reaction for basement membrane components (in amnios basement membrane and cells of umbilical cord stroma) has been added at the final of the manuscript.

(3) In figure 2 and 3, please correct abbreviations in the figure itself and the supporting figure text (e.g. SA / SAM, WH (WJ, etc.).

R/: Thank you for your helpful comments. Errors have been corrected to be more consistent with nomenclature throughout the manuscript.

(4) When providing quantitative data (e.g. in Fig. 4), please indicate which values are shown (mean, median, etc. with standard deviation or standard error, or other).

R/: Thank you for your helpful review. Type of values are now indicated in the last version of the manuscript.

(5) Please carefully read and revise or rewrite several sentences marked in yellow in the pdf document provided with this revision letter.

R/: It has been impossible to address the Reviewer request in this section due to the non-correspondence of the pdf file provided with our manuscript. We will be pleased to review the pertinent changes when we have the correct document.

Reviewer 2 Report

Overall, this article provides a comprehensive overview using detailed methodology, the characterization of the components of the WJ-BM and demonstrating the ability of stem cells derived from the WJ to create a native ECM microenvironment in an ex vivo setting. The discussion is very well structured and provides the reader with necessary references and context to the use of methodologies and outcome measures. 

1) What are the authors thoughts on using WJ-SCs with ECM extracted from WJ and what are the implications for the differentiation of the cells? 

2) What are the authors thoughts on using a concoction of generic basement membrane proteins (eg Matrigel) and how that would affect WJ-SC differentiation and microtissue formation? 

Author Response

Overall, this article provides a comprehensive overview using detailed methodology, the characterization of the components of the WJ-BM and demonstrating the ability of stem cells derived from the WJ to create a native ECM microenvironment in an ex vivo setting. The discussion is very well structured and provides the reader with necessary references and context to the use of methodologies and outcome measures. 

R/: We appreciate the reviewer's feedback and positive comments on our manuscript.

1) What are the authors thoughts on using WJ-SCs with ECM extracted from WJ and what are the implications for the differentiation of the cells? 

R/: The use of cord-extracted WJSCs that present these ECM components may be an interesting alternative for their differentiation to different cell lines that naturally present these compounds, such as chondrocytes or Schwann cells. These compounds have been shown to be crucial in the maintenance and growth of these cells, so the expression of these components previously by the cells may be an advantage for this purpose. Part of this information was included within the discussion of the manuscript (page 15, line 3-19).

2) What are the authors thoughts on using a concoction of generic basement membrane proteins (eg Matrigel) and how that would affect WJ-SC differentiation and microtissue formation?

R/: Matrigel has been shown to improve cell adhesion with respect to other substrates of different type, and can promote cell differentiation and serve as a scaffold to generate constructs of various dimensions. However, in the present study we addressed the generation of cell aggregates without the need for external scaffolds, promoting cell-cell adhesion and the creation of their own matrix. In this way, we originate microtissues formed exclusively by cells and the components of the extracellular matrix that they themselves synthesize. As the reviewer cleverly points out, it will be interesting to evaluate the behavior of microtissue generated here in addition to scaffolds as Matrigel to determine its proliferation rate and within different differentiation strategies.

Reviewer 3 Report

The manuscript is interesting.

The research question and the topic falls within the special issue aim and scope that have been selected. 

Importantly, with a tissue engineering perspective, the authors concluded that WJSC can synthesize most of the main BM ECM molecules ex vivo, especially when MT technique is used, in addition to provide new marker insights in WJ phenotype and stromal reconstruction of stem cell niche.

Abstract length is respected, following the instructions for authors of the Journal. 

The images are OK and legends clear. Material and methods sufficient to reproduce the experiments.  

I only have some comments, provided in the attempt to improve quality of this already high-quality article. 

Major comments

1)    At least a Graphical representation of percentage of positive cells and positive signal intensity, similar to Fig 4 for Fig 3, should be added for Figure 6.   

2)    Few days ago, a research group reported the ECM composition of spheroids made of WJSCs and these cultures appeared to have a condensed core and morphological similarities with MTs. It would be nice to make the paper more updated and give a comprehensive information of all WJ 3D cell culture if the authors will cite this paper, https://doi.org/10.3390/bioengineering10020189.

Minor comments

3)    Fig 1 legend, would you like to add “UC” definition as umbilical cord in the third sentence, close to other abbreviation description? 

4)    The performed histology staining (HE, AB, ect. ) of the umbilical cord is not providing any new information, but is OK to be shown for comparative analyses of the results derived from MTs and in vitro cultures of WJSCs, a short note mentioning the limitation and the non-novelty of the data presented in Figure 2 should be relevant. 

Author Response

The manuscript is interesting.

The research question and the topic falls within the special issue aim and scope that have been selected. 

Importantly, with a tissue engineering perspective, the authors concluded that WJSC can synthesize most of the main BM ECM molecules ex vivo, especially when MT technique is used, in addition to provide new marker insights in WJ phenotype and stromal reconstruction of stem cell niche.

Abstract length is respected, following the instructions for authors of the Journal. 

The images are OK and legends clear. Material and methods sufficient to reproduce the experiments.  

I only have some comments, provided in the attempt to improve quality of this already high-quality article. 

R/: Dear reviewer, thanks you very much for your kind and positive comments on the manuscript. We hope that the changes introduced will improve the present manuscript.

Major comments

1)    At least a Graphical representation of percentage of positive cells and positive signal intensity, similar to Fig 4 for Fig 3, should be added for Figure 6.   

R/: Thank for your interesting comment. It is an interesting datum, but due to three-dimensional structure of microtissues and especially the non-defined expression pattern of most basement membrane components within the MT, the quantification of the percentage of positive cells and positive signal intensity cannot be efficiently addressed.

2)    Few days ago, a research group reported the ECM composition of spheroids made of WJSCs and these cultures appeared to have a condensed core and morphological similarities with MTs. It would be nice to make the paper more updated and give a comprehensive information of all WJ 3D cell culture if the authors will cite this paper, https://doi.org/10.3390/bioengineering10020189.

R/: Thank you for pointing out the work not included in the article. Due to its novelty, it had not been published at the time the manuscript was written. It is now included in the references of the latest version of the manuscript as suggested (Page 3, line 2).

Minor comments

3)    Fig 1 legend, would you like to add “UC” definition as umbilical cord in the third sentence, close to other abbreviation description? 

R/: Thank you for your helpful comments. Abbreviation has been added to the latest version of the manuscript.

4)    The performed histology staining (HE, AB, ect. ) of the umbilical cord is not providing any new information, but is OK to be shown for comparative analyses of the results derived from MTs and in vitro cultures of WJSCs, a short note mentioning the limitation and the non-novelty of the data presented in Figure 2 should be relevant. 

R/: Dear reviewer, we understand your suggestion. It is truth that the histochemical methods were used within previous works (by our group), but the analysis of the findings was in a different context and aims. In the page 7 and Line 24 we stated the following sentence in order to clarify this information: ‘’Please note that the histochemical methods used to characterize main ECM molecules (AB, PAS and PS), have been used in previous works’’.